# Cervical cancer screening utilization and associated factors among female health workers in public health facilities of Hossana town, southern Ethiopia: A mixed method approach

**Zemzem Jemal[1], Nana Chea[2], Habtamu Hasen**  **[3]\*, Tsegaab Tesfaye[3], Netsanet Abera[2]**

**1** Department of Midwifery, Hossana College of Health Science, Hossana, Ethiopia, **2** College of Medicine and Health Sciences, School of Public Health, Hawassa University, Hawassa, Ethiopia, **3** Department of Public health, Hossana College of Health Science, Hossana, Ethiopia

\* habtamu130@gmail.com

## Abstract

### Backgrounds

Worldwide, a substantial proportion of women have low cervical cancer screening services utilization. There is a paucity of evidence in utilization of cervical cancer screening services among female health workers and inconsistent findings in Ethiopia. This study aimed to assess the utilization of cervical cancer screening services and associated factors among female health workers in public health facilities of Hossana town, Southern Ethiopia.

### Methods

Facility-based cross-sectional study design complemented with the qualitative inquiry was conducted among randomly selected 241 study participants in Hossana town from June 1 to July 1, 2021. Logistic regression models were used to determine the association between dependent and independent variables with the assumption of a variable with a p-value < 0.05 was considered statistically significant. Qualitative data were transcribed verbatim then translated to English and analyzed using open code version 4.03.

### Results

Out of the total study participants, 19.6% was screened for cervical cancer. Having a diploma level of education (AOR = 0.48;95%CI:0.24,0.98), having three or more children (AOR = 3.65;95%CI:1.44,9.21), having multiple sexual partners(AOR = 3.89;95%CI: 1.38,11.01), and knowledge of cervical cancer screening (AOR = 2.66;95% CI:1.19,5.95) were statistically significantly associated with cervical cancer screening utilization. In-depth interviews suggested additional barriers for low screening utilization including lack of health educational materials, limitation of service to a specific area, service interruption, provider incompetency, and miss-trust and lack of attention by a trained provider.

**Data Availability Statement:** All relevant data are within the paper and its Supporting Information files.

**Funding:** The authors received no specific funding for this work.

**Competing interests:** The authors have declared that no competing interests exist.

**Abbreviations:** AOR, Adjusted Odds ratio; CC, Cervical cancer; CI, Confidence interval; IDIs, Indepth interviews; HPV, Human Papillomavirus; STI, sexually transmitted infections; SD, Standard deviation; SRS, Simple random sampling; H/C, health center; WUNEMMCSH, Wachamo University Nigist Elleni Mohammed Memorial comprehensive Specialized Hospital.

## Conclusion

Utilization of cervical cancer screening service among female health workers is low. Having a diploma level of education, having three or more children, a history of multiple sexual partners, and knowledge about cervical cancer were predictors of cervical cancer screening utilization. Contextualized health talks and promotion through training with a special focus on low level of knowledge, had lower educational level, and the availability of cervical cancer screening services are critical.

## Introduction

The main cause of cervical cancer is Human Papilloma Virus (HPV) infection which is the most common viral infection of the reproductive tract, nearly all sexually active individuals will be infected with HPV at some point in their lives [1].

Cervical cancer (CC) is a serious public health problems and the fourth most common cancer detected among women globally. It is the second leading cause of female cancer-related deaths in sub-Saharan Africa [2, 3].

Each year, there were approximately 236,000 deaths from cervical cancer worldwide and it was the most common cancer in the east and middle Africa [1, 4]. About 90% of cases and 85% of these deaths have occurred in Low and Middle-Income Countries (LMICs); the highest has occurred in Sub-Saharan Africa (SSA) and approximately 311,000 women died from cervical cancer [2, 3].

Cervical cancer screening is watching for precursors before a person has any symptoms and has the benefit to reduce the incidence and the progression to an advanced stage of cancer as well as its mortality [5].

In Ethiopia, the utilization of cervical cancer screening is low and vary in different regions. For instance the prevalence of cervical cancer utilization in Arba Minch town, Southern Ethiopia (9.6%,) Sidama zone Southern Ethiopia(11.4%), and Mekelle town, northern Ethiopia (10.7%) among female health worker [6–8]. This implies that different contextual factors might have contributed to this different proportion of cervical cancer screening utilization among women in Ethiopia.

Different contextual factors that could have determined screening utilization have been identified for instance: limited access to information, lack of knowledge of cervical cancer, lack of healthcare infrastructure required, lack of trained practitioners, having multiple sexual partner, residence, and occupation [2, 6–9].

Global commitments to reduce the burden of cervical cancer have been done. For instance, the World Health Organization designed a 90–70–90 triple-intervention strategy aimed to achieve 90% HPV vaccination coverage, 70% of women being screened at least twice in their lifetime, and 90% of women having access to cervical pre-cancer and cervical cancer treatment and palliative care services by 2030 [10].

American obstetrics and gynaecology recommended that the initiation of screening with pap smear at the age of 21–29 years old every three years and HPV testing is taking every five years until the age of 65 [11]. Ethiopia also practiced cervical cancer screening method in health care facilities for women in the target group based on WHO recommendations [12].

Different studies have been conducted in Ethiopia to identify factors associated with cervical cancer screening utilization in different setting and in different population groups. Almost all studies have reached approximately the same conclusion: low utilization of cervical cancer

screening. However, most of these studies conducted cross-sectional study which inherently weak to identify unexplored contextual factors to the providers point of view.

We believe that a mixed approach of cross-sectional study would explore more about the contextual barriers for low utilization that measuring the same things using many quantitative approaches. This gaps calls further research works to identify hidden contextual factors using better study approaches to provide reliable evidences. Health workers are chief promotors of health care programs for their community, particularly; female health workers are role models that likely to have a better understanding of the benefits of cervical screening than others are and their utilization is a predictor of societal health behaviour on the control of cervical cancer [13].

Therefore, the purpose of this study was to assess cervical cancer screening services utilization and associated factors among female health workers.

## Methods and materials

### Study area

This study was conducted in Hossana town, the capital town of Hadiya Zone, found in the southern nation nationalities people's regional state (SNNPR) of Ethiopia. The town is situated 232 Km Southwest of Addis Ababa and 194 Km northwest of the regional town of Hawassa. It has an average elevation of 2276 meters above sea level and a total area of 23sq.km. The total population of Hossana town is 92,735 of this female accounts for 48808 and there are 21,607 reproductive age group women. In Hossana town administration there is one teaching Hospital, and three health centers namely Hossana health center, Bobicho health center, and Lichamba health center.

A total of 643 female health workers were in Hossana town public health facilities. For instance, nurses (271), midwives(113), laboratory technologists, and technicians(59),55 pharmacists and druggists(55), medical doctors(49), health officers(39), anaesthetics (13), 37 urban health extension workers(37), and five radiographers. Screening service for cervical cancer is giving in the teaching Hospital and Hossana health center [14].

### Study design, period, and population

We used a concurrent mixed method in which the implementation of qualitative and quantitative methods and merging the data to interpret the study results. The study was conducted from June 1 to July 1, 2021.

The source population comprised all female health workers working in the public health facilities of Hossana town, while the study population encompassed randomly selected female health workers in selected public health facilities of Hossana town. Female health workers were a study unit. Female health workers who were working in selected public health facilities of Hossana town and whose age is 21 and above were included in this study. Critically ill female health workers who were unable to communicate and female health workers on leave during data collection time were excluded from the study.

### Sample size determination and sampling technique

The sample size was calculated by applying two population proportion formulas using Epi-Info version 7 and taking a 5% margin error, 80% power, and a 1:1 ratio of an exposed group to a non-exposed group (r = 1). Assuming the proportion of attitude towards screening of cervical cancer (85.9%) and (AOR = 3.42) from a study conducted in Mekelle Town [8]. The calculated sample size was 332.

Since study population was small need correction formula, nf = n/1+ n/N

$$nf = 332/1 + 332/643 = 219$$

Considering the non-response rate of 10% in the estimation of the sample size required for the study. The final sample size for this study was 241.

For quantitative study, we used simple random sampling technique was used to recruit study participants. In Hossana town, four public health facilities (one comprehensive specialized hospital & three health centers). In Hossana town, four public health facilities (one comprehensive specialized hospital & three health centers) were available.

In the first stage, a sampling frame was prepared from the payroll of the human resources department in each four public health facility in Hossana town to determine the total numbers of female health workers. In the second stage, a total of 241 female workers were selected with probability proportion to the size of each health facility. Finally, the study participants were recruited using a simple random sampling method from each facility (Fig 1).

## Data collection tool and procedure

Data were collected using a structured and pretested questionnaire adapted in the English language from the available relevant literatures [6–8]. The questionnaires contained five parts,

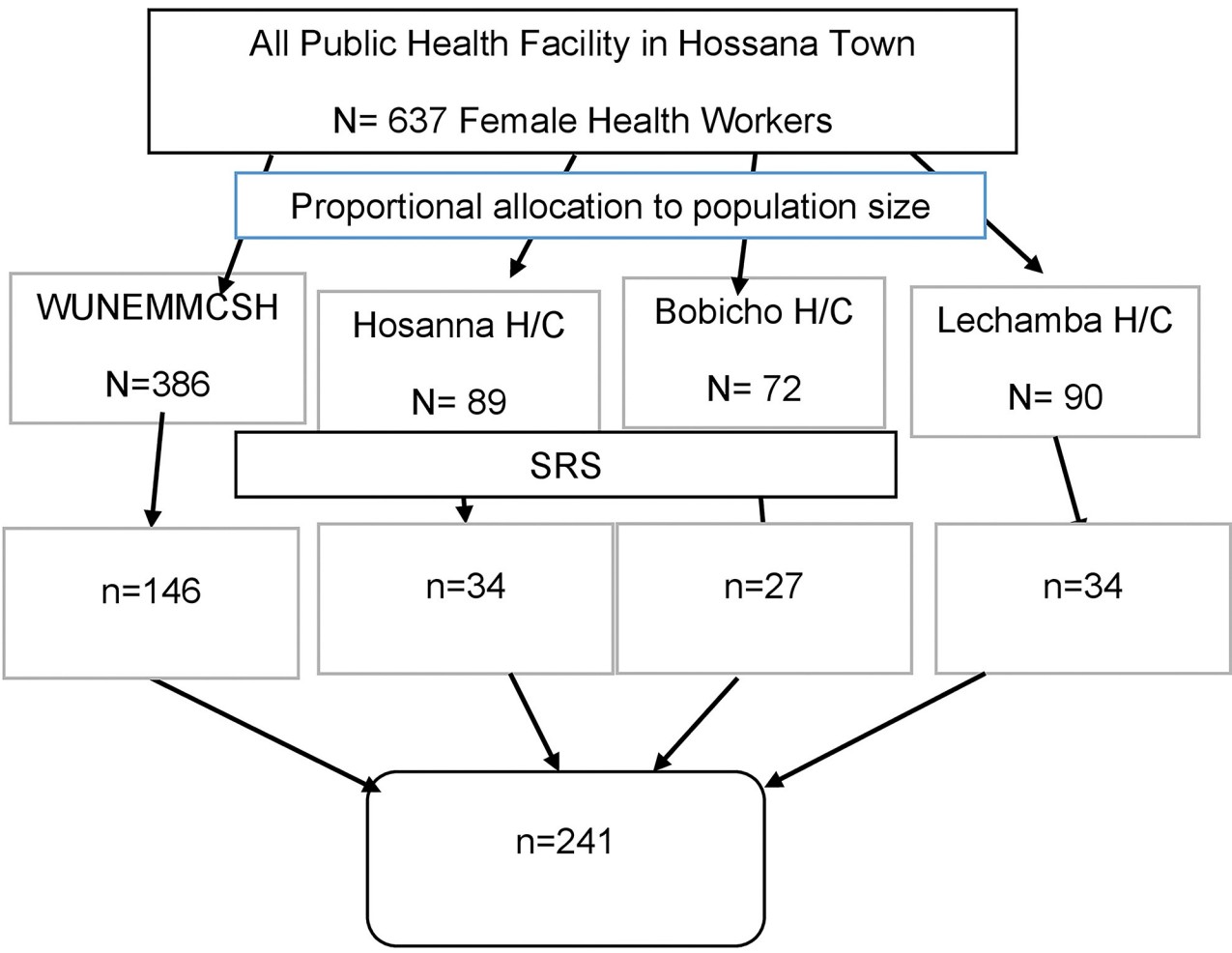

**Fig 1. Schematic presentation of the sampling procedure of this study.**

socio-demographic factors, knowledge of cervical cancer and screening-related factors, attitude/perception about cervical cancer and screening, and reproductive health and behavioral factors. The qualitative in-depth interview (IDIs) were conducted using a semi-structured interview guide to gain a deeper understanding of the participants to explore the barriers to utilization of cervical cancer screening. IDIs were administered to a purposely-selected subset of selected female health workers working in the cervical cancer-screening unit, who have management experience in leading the cervical cancer unit, have led the cervical cancer unit of town health administration, and female health workers in Hossana town public health facility. The interviews were completed at a time and place to suit the study participants and lasted between 20 and 45 minutes. Collecting data were stopped when IDIs participants fail to provide new information.IDIs were audio-recorded, translated from the local language into English, and then transcribed verbatim. Memos were used to understand contexts during interview.

## Operational definitions

**Female health workers.** Female health professionals who have contact with patients/clients including nurses, doctors, health officers, lab technicians/technologists, pharmacists/druggists, anesthetics, radiographers, urban health extension workers, public nurses, psychiatry nurses, and environmental nurses [6].

**Utilization of cervical screening.** Women screened at least once for cervical cancer in their life time [8].

**Knowledge about cervical cancer screening.** Was assessed using ten questions asked on knowledge of cervical cancer screening (risk factors about cervical cancer, prevention method, vulnerability to cervical cancer and symptoms of cervical cancer, frequency of screening, and screening method of cervical cancer). The response to each of the questions was "yes" or "no". Each correct answer was given a score of 1 while an incorrect answer was given a score of 0. We obtained composite knowledge ranging from 0 to 10 points. The scores from all 10 items were summed up and the mean sums of total scores were calculated. A female health worker who obtained scores of the mean and greater than the mean score was considered to have good knowledge and who obtained less than the mean score were considered to have poor knowledge [15].

**Attitude/perception of female health workers.** Towards cervical cancer screening was assessed using a likers scale which ranges from score five (strongly agree) to score one (strongly disagree). The responses were summed and a total score was obtained. Then we calculated the mean score. Those who scored the mean score and above were considered as having a favourable attitude or otherwise unfavourable attitudes toward cervical cancer screening [15].

## Measurements

The outcome variable, the utilization of cervical cancer screening, was measured through female health worker responses about where she ever screened at least once for cervical cancer.

## Data quality assurance

The questionnaires were translated to Amharic and then back-translated to English to assure the quality of data. Three-day training for data collectors and supervisors was given and the questionnaire was pretested in 10% of the study population in a different setting with a similar population in Fonko health center. Cronbach's alpha was done to assess internal consistency (alpha coefficient for knowledge on CCA (10 items) = 0.76, attitude on CCA screening (9 items) = 0.71.

Interview guides were prepared in the English language by language experts for qualitative study. Interviews were held in silent places which is suitable and comfortable for discussions. The audio recorder was checked for functionality before recording. During an interview the respondent's own words and crosschecked with notebooks. The recorded voice of interviews and notebooks were crosschecked while transcribing to ensure the credibility of the data.

## Data processing and analysis

The quantitative data were entered into Epidata version 3.1 and transported to SPSS version 23 software for analysis. Data were edited and cleaned by running a simple frequency, cross-tabulations, and sorting to identify outliers. Descriptive statistics like frequencies, percentages, and cross-tabulations were done. Binary logistic regression was used to check the associations of independent variables and outcome variables. Variables with p-values < 0.25 in the bivariable analysis were entered into multivariable analysis to isolate predictors. The goodness of fit of the model was checked using the Hosmer Lemeshow test of goodness of fit and variance inflation factors were low (<10) for the multi-collinearity check. An adjusted odds ratio with 95% confidence intervals and a p-value less than 0.05 were considered a statistically significant association with of utilization of cervical cancer screening.

All the qualitative data were systematically coded and analyzed using thematic analysis in open code 4.03 software. The audio recorder was transcribed verbatim in Amharic and then translated into English. The initial analysis was done by importing transcribed interviews to notepad and then again imported to open code 4.03 software. Starting from reading several times coding was performed line by line. After checking for similar groups of code were summarized into a category and final themes were created.

Categorizing and theming procedures were cross-checked by the other investigators and agreed on common categories and themes. We invited an expert to put sample of codes and categories to the emerged corresponding categories and themes for triangulation, respectively.

## Ethics approval and consent to participate

This study was approved by the Hawassa University College of medicine, and health sciences research ethics review committee (reference no. IRB/ 153/13). Written and signed informed consent were obtained from each study participant and head health facilities before the interview. The data collection procedure was anonymous to keep the confidentiality of any information provided by the study participants.

## Results

### Socio-demographic characteristics of study participants

A total of 235 female health workers participated in this study with a response rate of 97.5%. The mean (± SD) age of study participants was 28.8 (±4.94) years. About 163 (69.4%) of study participants were Hadiya in ethnicity and 178 (75.7%) were protestant religion followers. Concerning their educational status, two third (66%) were degrees and above. One hundred fifty-three (65.1%) of study participants had three and more years of working experience. Nearly three fourth (72.8%) of study participants were married and 142 (60.4%) were working in hospitals. Regarding profession 109(46.4%) of respondents were nurses (**Table 1**).

**Reproductive and behavioral characteristics.** One hundred seventy (72.3%) of study participants had their first sexual intercourse at an age greater than 18 years. Study participants who had a history of multiple sexual partners were 19 (8.1%) and 230 (97.9%) had no history

**Table 1. Socio-demographic characteristics of study participants in Hossana town, southern Ethiopia, 2021 (n = 235).**

| Variables | Categories | Frequency | Percent (%) |
|---|---|---|---|
| Age in years | ≤24 | 40 | 17.0 |
| | 25–34 | 159 | 67.7 |
| | ≥35 | 36 | 15.3 |
| Marital status | Married | 171 | 72.8 |
| | Single | 64 | 27.2 |
| Educational status | Diploma | 80 | 34.0 |
| | Degree and above | 155 | 66.0 |
| Religion | Protestant | 178 | 75.7 |
| | Muslim | 13 | 5.5 |
| | Orthodox | 41 | 17.4 |
| | Catholic | 3 | 1.3 |
| Service area | Hospital | 142 | 60.4 |
| | Health center | 93 | 39.6 |
| Working experience | ≤2years | 82 | 34.9 |
| | ≥3 years | 153 | 65.1 |
| Professions | Nurse | 109 | 46.4 |
| | Health officer | 29 | 12.3 |
| | Doctors | 12 | 5.1 |
| | Pharmacy | 12 | 5.1 |
| | Laboratory | 26 | 11.1 |
| | Midwifery | 34 | 14.5 |
| | Others* | 13 | 5.5 |

* = radiologist, anesthesia

of sexually transmitted disease(STDs). Eighty-nine (37.9%) of respondents were nulliparous women and nearly all (98.8%) of the respondents never smoked (**Table 2**).

**Female health worker's knowledge of cervical cancer.** One hundred fifty-one (64.3%) of study participants had good knowledge about cervical cancer screening. Regarding symptom-related knowledge, more than half (51.5%) of study participants mentioned risk factors for cervical cancer by respondents were having multiple sexual partners, early sexual intercourse 106

**Table 2. Reproductive and behavioral characteristics of study participants in Hossana town southern Ethiopia, 2021 (n = 235).**

| Variables | Categories | Frequency | Percent (%) |
|---|---|---|---|
| Parity status | Nulli parity | 89 | 37.9 |
| | 1–2 child | 83 | 35.3 |
| | ≥ 3 child | 63 | 26.8 |
| Age at first sexual intercourse | ≤ 18year | 65 | 27.7 |
| | > 18 year | 170 | 72.3 |
| Having multiple sexual partners | Yes | 19 | 8.1 |
| | No | 216 | 91.9 |
| History of STDs | Yes | 5 | 2.1 |
| | No | 230 | 97.9 |
| Smoking status | Yes | 3 | 1.3 |
| | No | 232 | 98.8 |

Table 3. Knowledge of risk factors, symptoms, and screening methods among female health workers of Hossana town southern Ethiopian 2021 (n = 235).

| Variables | | No of responded yes | Percentage (%) |
|---|---|---|---|
| Knowledge of Risk factors * | Having multiple sexual partners | 121 | 51.5 |
| | Early sexual intercourse | 106 | 45.1 |
| | Acquiring HPV virus | 107 | 45.5 |
| | Cigarette smoking | 50 | 21.3 |
| Knowledge of symptoms* | Vaginal bleeding | 143 | 60.9 |
| | Foul-smelling vaginal discharge | 143 | 60.9 |
| | Contact bleeding | 105 | 44.7 |
| | Postmenopausal bleeding | 49 | 20.9 |
| Knowledge of screening methods* | Pap smear | 108 | 46.0 |
| | VIA | 15 | 6.4 |
| | VILI | 45 | 19.1 |
| | HPV DNA test | 51 | 21.7 |

NB. Those with an asterisk (*) were not added up to 100% because of multiple responses

(45.1%), acquiring HPV virus 107 (45.5%), and Cigarette smoking 50 (21.3%). One hundred forty-three (60.9%) of study participants listed vaginal bleeding and foul-smelling vaginal discharge and contact bleeding accounts (44.7%) and postmenopausal bleeding 49 (20.9%) were the symptoms of cervical cancer.

Knowledge related to cervical cancer screening method were assessed and study participant mentioned pap smear108 (46.0%), HPV DNA test 51(21.7%), VILI 45(19.1%), and only 15 (6.4%) stated the VIA as the screening methods (**Table 3**).

Corresponding to the quantitative finding, key informant interview participants most frequently mentioned female health workers have a low understanding of cervical cancer screening services, as illustrated below:

*". . ..The root cause of the problems are health care workers not much understood about the disease and the availability of screening service, as I think there is no adequate understanding. However cervical cancer is known as a killer disease but there is no sufficient understanding about prevention methods, risk factors, and availability of screening services, as I think these all may reason for underutilization."* [Female, Age:34, Reproductive health specialist]

*Another participant reaffirmed the above saying ". . .Ok what makes female health workers for not being screened as I tell you above they have no knowledge and awareness and also do not know as the service present in the facility. The first thing they have no specific knowledge about cervical cancer that makes them screen"* [Female, Age: 26 midwifery].

**Attitude towards cervical cancer screening.** More than half of the respondents (51.5%) had a favourable attitude towards cervical cancer screening. As shown in the Table below, 80.4% of study participants perceived that cervical cancer is the killer cancer and 80% of study participants perceived that cervical cancer screening helps in the prevention of carcinoma of the cervix. Out of the total respondents, 177(75.3%) agree that Cervical cancer screening causes no harm to the clients, and 171(72.8) participants will screen for cervical cancer if the service needs payment. The majority of the participants 188 (80%) agreed that cervical cancer screening tests find changes before it becomes cervical cancer (**Table 4**).

**Table 4. Attitude towards cervical cancer screening among female health workers in Hossana town, southern Ethiopia, 2021 (n = 235).**

| Variables | Level of agreements | |
|---|---|---|
| | Agree Frequency (%) | Disagree Frequency (%) |
| CC is killer cancer in Ethiopia? | 189(80.4) | 46(19.6) |
| CC screening helps in prevention of carcinoma of cervix? | 188(80) | 47(20) |
| CC screening causes no harm to the clients | 177(75.3) | 58(24.7) |
| I will screened for cervical cancer if the service need payment | 171(72.8) | 64(27.2) |
| Adult women including you could be acquired cervical cancer | 184 (78.3) | 51(21.7) |
| CC screening tests find changes before it becomes cervical cancer | 188 (80.0) | 47(20.0) |
| CC screening procedure is embracement | 38 (16.2) | 197(83.8) |
| CC screening will you allow male doctors to examine you | 155 (66.0) | 80(34.0) |
| If you have cervical cancer do you consult doctors without being scarce | 217 (92.3) | 18(7.7) |

## Barriers for cervical cancer screening service utilization

The most common barriers for cervical cancer screening utilization in health facilities were study participants assume themselves as healthy (54.8%), do not know the place of service (11.5%) and carelessness (11.1%) on screening method(**Fig 2**)

Corroborating to our quantitative findings,

Two IDIs interviewees pointed out many barriers that may prevent female health workers from using screening services because of assuming a healthy; do not know where service is given in their facility, fear of pain of screening procedure, and being Carelessness as illustrated below:

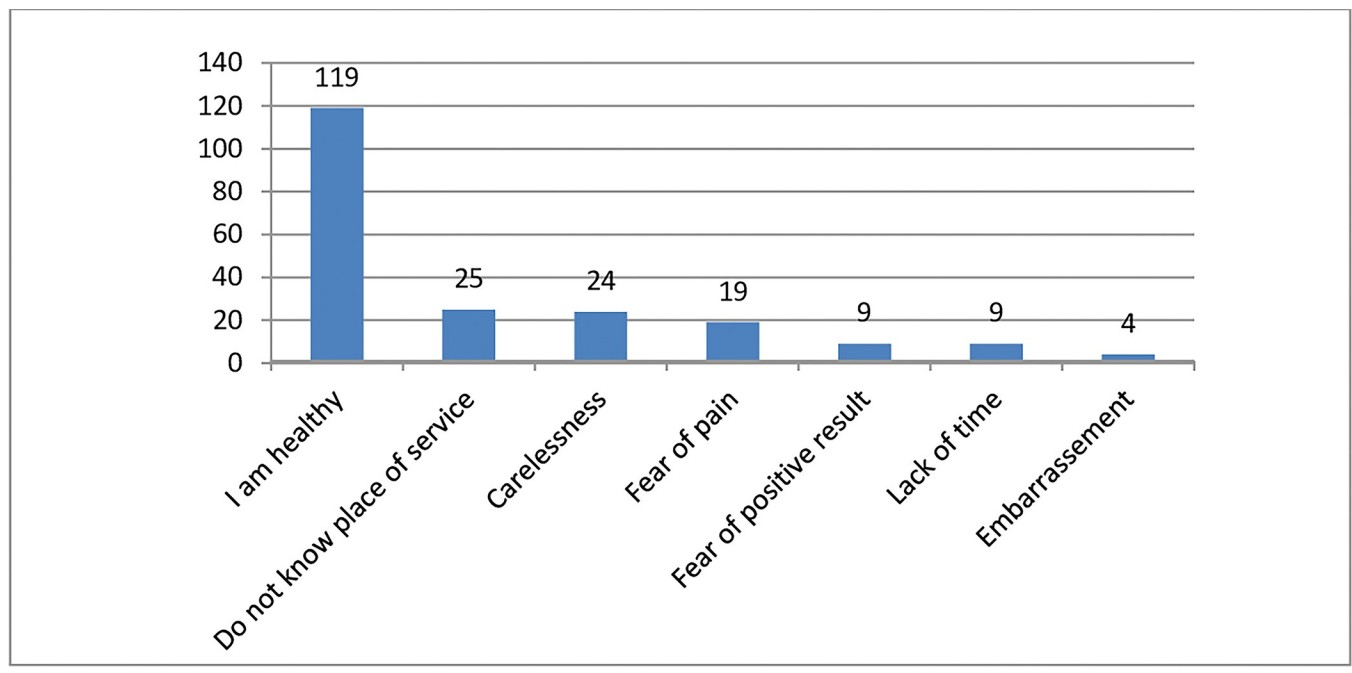

**Fig 2. Barriers for cervical cancer screening utilizations among female health workers in Hossana Town, Southern Ethiopia, 2021.**

*"...a problem is we always think that as we always live healthily. I think the prevalence is increasing currently by understanding these, everyone must be aware as they may acquire the disease I need to tell to all mothers to internalize this idea" (Age: 29, female health workers).*

Another key informant indicated that carelessness and knowledge-related factors play important role in getting screened saying

*"....As I think what makes female health worker not to be screened is carelessness, lack of attention and knowledge...." [Female, Age: 26, midwifery working in screening unit].*

Around one-third (32.3%) of study, participants mentioned that cervical screening service is not convenient with their regular working time.

In line with this finding, results from a qualitative study showed that female health workers regular working time were not convenient with the screening time schedules. Key informant participants mentioned the barrier for low utilization screening services is they are busy and overburdened with their duty, so they have no time to screen. Therefore arranging a screening on weekend days is necessary. A 26-year-old participant said: "*Indeed female health workers most of the time are busy due to their duty associated with their work. So they have no extra time to screen, therefore arranging to screen on weekend days is necessary to increase cervical cancer screening utilization.*" [Female, Age: 36, midwifery working in screening unit].

## Cervical cancer screening service utilization

According to the finding of this study, forty-six (19.6%, 95% CI: 14.5%, 24.7%) of the study participants had ever been screened at least once for cervical cancer.

## Factors associated with utilization of cervical cancer screening service

Table 5 summarizes a bivariate logistic regression analysis of socio-demographic and other characteristics of study participants that are associated with female health workers' utilization of cervical cancer screening services. Among those variables, age, marital status, educational status, service area, working experience, history of multiple sexual partners, parity, comprehensive knowledge of cervical cancer and its screening method were positively associated with female health worker's utilization of cervical cancer screening services during bivariate analysis (**Table 5**).

The odds of cervical cancer screening service utilization were 3.65 times [(AOR = 3.65; 95% CI: (1.44, 9.21)] higher among female health workers having three or more children as compared to those who were nulliparous female health workers. The odds of cervical cancer screening service utilization among diploma female health workers were 52% less likely [(AOR = 0.48; 95% CI: (0.24, 0.98)] when compared with those who were degree and above educational status.

The key informant explained that female health workers with low-level educational status had a low-level understanding so that not utilize the service, saying:

*"...When looked at while I provide training, most of the health workers especially those with low educational levels have difficulty with the understanding of cervical cancer screening. Now the problem among health workers a little bite knowledgeable put them in unnecessary confidence resulted in them for the resistant of screening."[Male, Age: 45, gynaecologist].*

**Table 5. Multivariable logistic regression analysis on factors associated with utilization of cervical cancer screening among female health workers in Hossana town, southern Ethiopia, 2021 (n = 235).**

| Variables | Utilization of cervical cancer screening | | COR (95%CI) | AOR (95%CI) |
|---|---|---|---|---|
| | Yes n (%) | No n (%) | | |
| Age in year | | | | |
| ≤24 | 4 (10.0) | 36 (90.0) | 1 | 1 |
| 25–34 | 30 (18.9) | 129 (81.1) | 2.09(0.69,6.33) | 1.22 (0.34,4.28) |
| ≥35 | 12 (33.3) | 24 (66.7) | 4.50(1.21,5.61) | 1.38 (0.31,6.33) |
| Marital status | | | | |
| Married | 40 (23.4) | 131(76.6) | 2.95(1.18,7.35) | 1.64(0.45,6.01) |
| Single | 6 (9.4) | 58 (90.6) | 1 | 1 |
| Service area | | | | |
| Hospital | 33 (23.2) | 109(76.8) | 1.86(0.92,3.76) | 1.82 (0.88, 4.05) |
| Health center | 13 (14.0) | 80(86.0) | 1 | 1 |
| Parity status | | | | |
| Nulli parity | 9(10.1) | 80(89.9) | 1 | 1 |
| 1–2 Children | 18(21.7) | 65(78.3) | 2.46(1.04,5.84) | 2.15(0.87,5.31) |
| ≥3 children | 19(30.2) | 44(69.8) | 3.84(1.60,9.20) | 3.65(1.44,9.21)** |
| Educational status | | | | |
| Diploma | 24(15.5) | 131(84.5) | 0.48(0.25,0.93) | 0.48 (0.24,0.98)** |
| Degree and above | 22(27.5) | 58(72.5) | 1 | 1 |
| Working experience | | | | |
| ≤2 year | 9(11.0) | 73(89.0) | 1 | 1 |
| ≥3 year | 37(24.2) | 116(75.8) | 2.58(1.18,5.67) | 1.52(0.57,4.03) |
| Knowledge of cervical cancer screening | | | | |
| Good | 36(23.8) | 115(76.2) | 2.32(1.08,4.95) | 2.66(1.19,5.95)** |
| Poor | 10(11.9) | 74(88.1) | 1 | 1 |
| Had multiple sexual partners | | | | |
| Yes | 8(42.1) | 11(57.9) | 3.41(1.28,9.04) | 3.89(1.38,11.01)** |
| No | 38(17.6) | 178(82.4) | 1 | 1 |

** = significant variables at p value level <0.05  1 = reference group

Concerning knowledge-related factors, the odds of cervical cancer screening service utilization were 2.66 times (AOR = 2.66;95% CI: 1.19,5.95) higher among female health workers who had good knowledge of cervical cancer screening, risk factors, and symptoms as compared to who had poor knowledge.

Of nine interviews, six of them said poor knowledge and lack of awareness are the main reason for not utilizing screening services, as illustrated below:

Another key informant mentioned even those who had awareness were not properly using cervical cancer screening services, as illustrated below:

*"Female health workers did not give attention for cervical cancer screening as a general and they do not know means as there is no anything expose me, they not feel about screening whether they have screened or not screened."[A Female, Age: 28 BSc midwifery].*

Similarly, the odds of cervical cancer screening service utilization were 3.89 times (AOR = 3.89 95% CI: 1.38, 11.01) higher among female health workers who had a history

of multiple sexual partners as compared to those not having multiple sexual partner histories.

## Discussion

This study aimed to assess cervical cancer screening utilization and associated factors among female health workers in Hossana town public health facilities. The utilization of cervical cancer screening services in this study was 19.6%. The study further revealed that factors like education level, parity, history of multiple sexual partners, and knowledge of cervical cancer were significantly associated with the utilization of cervical cancer screening services.

In this study, the finding of cervical cancer screening services utilization was 19.6%. This is congruent with the studies conducted in Ethiopia (22%), in lower resource settings of Nigeria (20.6%), Baghdadi (18.8%), Chennai corporation (18.4%), Tanzania (15.4%), and in Dar es Salaam, Tanzania (21%) [16–21]. This consistency could be comparable with the current socio-demographic status and sample size. However. This study was lower than studies done in Saudi Arabia (26.2%), Uganda (75%), Cameroon (41%), and Ibadan, Nigeria (34.6%) [15, 22–24]. This difference might be due to the time of implementing cervical cancer screening practice and the difference in the level of countries' health service coverage. The finding of this study also higher than the studies done in Sidama zone, Southern Ethiopia (11.4%), Arba Minch town, Southern Ethiopia (9.6%), Mekelle town, northern Ethiopia (10.7%), Uttar Pradesh, India (10%), Korea (13%), South-eastern Nigeria (7.2%), rural India (7%), Sokoto, Nigeria (10%) [6–8, 25–29]. This inconsistency may be due to time variation, and differences in the study setting. In addition, our study participants were mainly female health workers living in urban setting which probably accounted for the observed difference.

The most common barrier for low utilization of cervical cancer screening services were participants assume themselves as being healthy (54.8%), do not know where the service provided (11.5%) and carelessness (11.1%). These misconceptions need to be in an intervention programs targeting this category of health workers. This finding is supported by studies conducted in Korea and Arba Minch town, Southern Ethiopia [6, 26]. In addition, when people are feeling and assumes themselves as being healthy they do not bother about preventive services as they have other competing problems. This implies that study participants who do not know the service area of cervical cancer screening further fuel the underutilization of screening services.

Our qualitative analysis showed that various challenges exist in the screening journey for female health workers women in the study area. Amid the numerous barriers; service users factors (feeling of being healthy, do not know what service is given in their facility, fear of pain, and carelessness). Furthermore, participants explained as lack of health educational materials, not having appropriate supplies and logistics, limitation of service to specific area, service interruption.

Service Provider's factors like: provider incompetency, miss-trust, lack of attention by a trained provider, and unsuitability of environment hindered female health worker from the utilization of cervical cancer screening.

Educational status was one of the significant factors in the utilization of cervical cancer screening services. Diploma female health workers were less likely to utilize cervical cancer screening services compared with those who had a degree and above educational status. This finding was supported by the studies done in Debremarkos town in Northwest Ethiopia, Wolaita zone, Southern Ethiopia and Nigeria [29–31]. This implies that as peoples experienced with more education, they have opportunity to understand the cause, risk factors, prevention mechanism, and screening methods of cervical cancer. Furthermore, education can increase

access to information from different sources within their educational career and have a positive effect on self-efficacy, confidence, and motivation, in search of health interventions for their health.

The odds of cervical cancer screening service utilization were higher among female health workers having three or more children as compared to nulliparous. The result of our study is supported with studies conducted in low resource setting areas of Nigeria, rural India, and Debremarkos Northwest Ethiopia [20, 28, 30]. This result indicated that female health workers with three and more children would have experienced repeated exposure to different contact in different service area of health facilities. History of multiple sexual partners is also an important predictor of cervical cancer screening utilization. The odds of cervical cancer screening service utilization were higher among female health workers who had a history of multiple sexual partners as compared to counterpart. Consistent studies conducted in the Tigray region, Northern Ethiopia, Debremarkos town, Northwest Ethiopia, and Addis Ababa, Ethiopia [8, 30, 32]. The possible mechanism might be the more sexual partners a woman has, the greater her chances of becoming infected with the human immune-deficient virus and other sexually transmitted diseases including Human Papillomavirus, the most common risk factor for the development of cervical cancer. The other justification might be perceived fear of getting sexually transmitted disease, which increased health facility visits, and the chance of seeking medical help. Similarly, this study revealed that female health workers' knowledge level has a positive effect on cervical cancer screening utilization. The odds of cervical cancer screening service utilization were higher among female health workers who had good knowledge of cervical cancer screening, risk factors, symptoms, and screening methods as compared to those who had poor knowledge. Similar study done in Arba Minch town, Southern Ethiopia supported our findings [6]. As female health workers have good knowledge about cervical cancer, they do not missed the opportunity to increase early detection and improve their survival.

The limitation of this study is the fact that since the study design was a cross-sectional study, temporal relations could not be established. In addition to this, since it is a facility-based study and included participants only from selected health institutions, it does not consider women who did not visit and not working in the health facilities. This might affect the representatives of our findings to Hossana town. The data were self-report by the study participants; thus subject to recall and social desirability bias may affect the result of the study. Regardless of these limitations, our findings have a strength of using the mixed quantitative and qualitative methods allowing for triangulation to confirm findings and qualitative data to address health facility-related factors, service provider-related factors, female health worker-related factors, and cultural factor inquiry for further explanation of ideas.

## Conclusion

This study revealed that utilization of cervical cancer screening service among female health workers is relatively lower than the Ethiopian national guideline for cervical cancer prevention and control. Educational status, parity of respondent, history of multiple sexual partners, and knowledge about cervical cancer were significantly associated with cervical cancer screening utilization. The common barriers for low utilization of screening services were feeling of healthy and screening time is not convenient.

Moreover, IDIs described that service user's (fear, carelessness and assume being healthy), service provider's (provider mistrust, shortage of trained man power and incompetency), health facilities factors (service interruption, lack of materials) were hindered from the utilization of cervical cancer screening.

Based on the findings of our study we recommend concerned bodies in the health sector to design strategic efforts aimed at improving female health work's knowledge on cervical cancer. It is also highly valuable to consider upgrading their educational level to realize improved screening service uptake among the groups. Moreover, contextualized health talks and promotion on the availability of screening services and maintain logistics are critical.

All contributed significantly and gave the final approval for the paper to be published; agreed to be accountable for all impacts of the work.

## Supporting information

**S1 Data. Data collection tool.**
(DOCX)

**S2 Data. SPSS data set.**
(SAV)

## Acknowledgments

We would like to extend our deepest gratitude to Hawasa University for ethical approval. Our appreciation also goes to the data collectors, supervisors, and study participants.

## Ethics approval and consent to participate

Permission to conduct the study was obtained from Hawassa University, College of medicine and health sciences research ethics review committee. Written and signed informed consent were obtained from each study participant and head health facilities before the interview. The data collection procedure was anonymous to keep the confidentiality of any information provided by the study participants.

## Author Contributions

**Conceptualization:** Habtamu Hasen, Netsanet Abera.

**Data curation:** Zemzem Jemal.

**Formal analysis:** Zemzem Jemal.

**Funding acquisition:** Zemzem Jemal.

**Investigation:** Zemzem Jemal.

**Methodology:** Zemzem Jemal, Nana Chea, Habtamu Hasen, Netsanet Abera.

**Resources:** Zemzem Jemal.

**Software:** Zemzem Jemal, Habtamu Hasen, Netsanet Abera.

**Supervision:** Nana Chea, Habtamu Hasen, Tsegaab Tesfaye.

**Writing – original draft:** Zemzem Jemal, Habtamu Hasen.

**Writing – review & editing:** Zemzem Jemal, Nana Chea, Habtamu Hasen, Tsegaab Tesfaye, Netsanet Abera.

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
