## [Decision Letter · Decision Letter 0]

22 Mar 2023

PONE-D-22-33914Cervical cancer screening utilization and associated factors among female health workers in public health facilities of Hosanna town, southern Ethiopia: A mixed method approachPLOS ONE

Dear Dr. Hassen,

Thank you for submitting your manuscript to PLOS ONE. After careful consideration, we feel that it has merit but does not fully meet PLOS ONE’s publication criteria as it currently stands. Therefore, we invite you to submit a revised version of the manuscript that addresses the points raised during the review process. Thank you for submitting this manuscript, dear authors. I tried to assess the entire manuscript, and it is very interesting. However, there have been many primary and systematic review studies on cervical cancer in Ethiopia.What is your specific drive for conducting this research?What is your new finding from this study that differs from previous research? Please respond  for all question and comments one by one

We look forward to receiving your revised manuscript.

Kind regards,

Gedefaye Nibret Mihretie, MSc

Academic Editor

PLOS ONE

Journal Requirements:

2. Our staff editors have determined that your manuscript is likely within the scope of our Early Detection, Screening and Diagnosis of Cancer Call for Papers. This editorial initiative is headed by in-house PLOS editors. This Call for Papers aims to explore recent advances in the early detection of cancer and implications of these advances for patient survival. Additional information can be found on our announcement page: https://collections.plos.org/call-for-papers/early-detection-screening-and-diagnosis-of-cancer/

If you would like your manuscript to be considered for this collection, please let us know in your cover letter and we will ensure that your paper is treated as if you were responding to this call. Please note that being considered for the Call for Papers does not require additional peer review beyond the journal’s standard process and will not delay the publication of your manuscript if it is accepted by PLOS ONE. If you would prefer to remove your manuscript from collection consideration, please specify this in the cover letter.

"No ,the funders had no role in study design, data collection and analysis, decision to publish, or preparation of the manuscript."

6. Please upload a copy of Supporting Information Figures 1 and 2 which you refer to in your text on page 32.

Reviewers' comments:

Reviewer's Responses to Questions

**Comments to the Author**

1. Is the manuscript technically sound, and do the data support the conclusions?

Reviewer #1: Partly

Reviewer #2: Partly

2. Has the statistical analysis been performed appropriately and rigorously? 

Reviewer #1: Yes

Reviewer #2: Yes

3. Have the authors made all data underlying the findings in their manuscript fully available?

Reviewer #1: Yes

Reviewer #2: No

4. Is the manuscript presented in an intelligible fashion and written in standard English?

Reviewer #1: Yes

Reviewer #2: No

5. Review Comments to the Author

Reviewer #1: What is new in your finding there are studies conducted in Ethiopia?

Your background section in abstract is too long. So try to minimize 3-4 sentences

Again your justification of study is not strong to conduct this study.

Your introduction is better in organizing like presenting in the first paragraph, a contextualization on cervical screening. In the second paragraph the difficulties and associated factors. In the third paragraph, the justification for carrying out the research, and in the last paragraph, finalize it with the research objectives

Your objective not clearly stated.

Reviewer #2: General Comments: Thank you for the manuscript. Any study on CC in LMIC country is important as efforts are made to control/eliminate CC globally. The manuscript has some merits but needs a lot more to be done to make it publishable.

First, as a general comment, you should get the whole manuscript edited by someone with English as the first language. The current version has significant issues with the use of past and present tenses, the absence of punctuation marks here needed. I have pointed some out below but get this fixed throughout the manuscript.

Specific comments

Introduction

1. Line 14: change “portion” to proportion and also you write that the women “had” low CC screening. My concern is with the use of the word “had” in the past tense without reference to a specific year or time. I suggest you write that they have low CC screening…

2. Line 17: health workers role in promotion positive health seeking behaviour including CC screening. The current statement suggest that their role is specific to CC screening

3. Lines 75-77: the sentence does not end in a meaningful way so do rephrase.

4. Lines 82-84: again, another sentence that is confusing. To check and rephrase. It currently state “However, none of these studies have assessed the association between the convenience of screening time and the barrier to utilizing cervical cancer screening services were not qualitatively captured”.

5. Line 84: to state that you aim to determine the “prevalence” in this context is not the most appropriate term. You could say you are determining service utilisation rate or something like that so please rephrase

6. Lines 86-89: this sentence should come just before the study aim.

Methos

7. Lines 100 – 104: rephrase and made the sentence clear and concise.

8. You need to provide more details on the study recruitment process and data collection.

o How exactly were the HCWs recruited for the quantitative arm?

o Was this a concurrent mixed method, or sequential or what?

o How were those HCWs recruited for the interviews for the qualitative arm? Who conducted the interviews?

o Where, for how long? Etc. etc.

o Since these were HCWs why were the interviews not conducted in English? What measures were in place to ensure that the translation process, did not affect the content of what was said? Was member checking done after transcription?

o Could someone who responded to the quantitative survey also be in the qualitative interviews?

o What informed the sample size for the qualitative interview?

9. Insert the ethical approval reference number.

Results

10. Why was the age ranges done in this manner? Why not start with the youngest age and then create the ranges to be meaningful?

11. Lines 250-251: what does “hugeness and hardiness” mean? If you say this was verbatim, I am wondering what the person said that is translated as hugeness and hardiness. It makes the sentence confusing to the reader. Since the statements after that by the same person sounds clearer, I suggest you delete this first sentence. In fact, the quotations all sound as if the people have low educational level but these are midwives and other level of trained health professionals. I believe the problem is trying to do direct translation from the local dialect to English without giving meaning to the statements. This does not help in this case to make the manuscript read well. Take a look again at your approach to the qualitative data analysis and presentation.

12. Rephrase the sentence at line 296 and 299 about CC service utilisation.

13. Lines 301-304: this whole part needs to be rephrased. The English language use is poor. This really needs to be addressed throughout the manuscript.

14. You repeat to many quotations about the level of CC knowledge, so you must review this and delete some to shorten the manuscript and also make it less boring to the reader.

15. Line 393 has no meaning now. Rephrase because I can’t think through eve to understand what you intend to say

16. Line 395-396: what is lower? Are you still talking about the utilisation rate? If so, be explicit.

Discussion and conclusions

17. These are also full of grammatical errors and these needs to be dealt with.

18. And then it should expand on recommendations which can address the barriers identified in this study.

6. PLOS authors have the option to publish the peer review history of their article (what does this mean?). If published, this will include your full peer review and any attached files.

Reviewer #1: No

Reviewer #2: No

---

## [Author Response · Author response to Decision Letter 0]

17 Apr 2023

Author’s response to reviews

Title: Cervical Cancer Screening Utilization and Associated Factors among Female Health Workers in Public Health Facilities of Hosanna Town, Southern Ethiopia: A Mixed Method

Approach

Authors:

Zemzem Jemal (zemzemj6@ gmail.com)

Nana Chea (cheanana2007@gmail.com)

Habtamu Hassen (habtamu130@gmail.com)

Tsegaab Tesfaye (habtamu130@gmail.com)

Netsanet Abera(netsaneta@hu.edu.et)

Version: 1 Date: 17 Apr 2023

Author’s response to reviews:

We thank the esteemed reviewer for their valuable comment. Your input is very much helpful in improving our work. We have addressed the comments of the reviewers point by point as follows, for clarity we have highlighted our response in the main text as well as here.

Academic Editor comments

#1Thank you for submitting this manuscript, dear authors. I tried to assess the entire manuscript, and it is very interesting. However, there have been many primary and systematic review studies on cervical cancer in Ethiopia.

Response: Thank you very much for your critical comments. We acknowledge that different studies were done. However, Almost all studies have reached approximately the same conclusion: low utilization of cervical cancer screening. However, most of these studies conducted cross-sectional study which inherently weak to identify unexplored contextual factors to the providers point of view.

#2 What is your specific drive for conducting this research?

The recent increase in cervical cancer disease will alarm the researchers to explore the root cause of the problems and researchers further addressed does health care workers utilized the initiatives of cervical cancer screening services?.

We believe that a mixed approach of cross-sectional study would explore more about the contextual barriers for low utilization that measuring the same things using many quantitative approaches. This gaps calls further research works to identify hidden contextual factors using better study approaches to provide reliable evidences. Health workers are chief promotors of health care programs for their community, particularly; female health workers are role models that likely to have a better understanding of the benefits of cervical screening than others are and their utilization is a predictor of societal health behavior on the control of cervical cancer

#3 What is your new finding from this study that differs from previous research? Please respond for all question and comments one by one

Response: Thank you very much for your critical comments. Yes, we believed it has new things

 Previous studies were mainly focused on quantitative study and focused on specific female health workers like Health extension workers and nurses. But, the current study was focused by including all female health workers (Nurses, health office, midwifery, pharmacy, general practitioners, anesthetists etc…). 

 Prior study was not considered some variables like convenience of screening time for female health workers in which they were already busy with the working hours was assessed?. Moreover, health facility factors (service interruption, lack of logistics) and provider factors (lack of training) were addressed in the current study.

 The uptake and scale-up of such initiatives of cervical cancer screening services approaches largely depend on knowledgeable, motivated and capable health-care providers that are well respected by the communities they serve. Moreover, health workers were a chief advocators of health services initiatives

 Majority of studies were focused on the community and health professionals were neglected because the researchers assumes that health-care providers were made aware of the workplace program at clinical meetings, public events, as well as study education and training events.

 The use of mixed methods approaches have become increasingly popular in recent years in the field of health sciences (Bressan V, Bagnasco A, Aleo G, et al. Mixed-methods research in nursing - a critical review. J Clin Nurs. 2017;26(19–20):2878–2890).

Reviewer 1

#1: What is new in your finding there are studies conducted in Ethiopia?

Response: Thank you very much for your critical comments. Yes, we believed it has new things

 Previous studies were mainly focused on quantitative study and focused on specific female health workers like Health extension workers and nurses. But, the current study was focused by including all female health workers (Nurses, health office, midwifery, pharmacy, general practitioners, anesthetists etc…). 

 Prior study was not considered some variables like convenience of screening time for female health workers in which they were already busy with the working hours was assessed?. Moreover, health facility factors (service interruption, lack of logistics) and provider factors (lack of training) were addressed in the current study.

 The uptake and scale-up of such initiatives of cervical cancer screening services approaches largely depend on knowledgeable, motivated and capable health-care providers that are well respected by the communities they serve. Moreover, health workers were a chief advocators of health services initiatives

 Majority of studies were focused on the community and health professionals were neglected because the researchers assumes that health-care providers were made aware of the workplace program at clinical meetings, public events, as well as study education and training events.

 The use of mixed methods approaches have become increasingly popular in recent years in the field of health sciences (Bressan V, Bagnasco A, Aleo G, et al. Mixed-methods research in nursing - a critical review. J Clin Nurs. 2017;26(19–20):2878–2890).

#2: Your background section in abstract is too long. So try to minimize 3-4 sentences

Response: Thank you very much for your critical comments. Comments were taken and revisions have been made accordingly.

#3: Your justification of study is not strong to conduct this study?.

Response: Thank you for your concern. Correction was made

#4: Your introduction is better in organizing like presenting in the first paragraph, a contextualization on cervical screening. In the second paragraph the difficulties and associated factors. In the third paragraph, the justification for carrying out the research, and in the last paragraph, finalize it with the research objectives

Response: Thank you for your critical comments. Comments were taken and revisions have been made accordingly. 

#5: Your objective not clearly stated.

Response: Thank you. Comments were taken accordingly. 

#6: is it the real data? In three health facilities much female workers?

Response: Thank you for your concern. Definitely, one of the three health facilities, one was a comprehensive specialized hospital with larger proportion of female workers. 

#7: Try to minimize the information in the study area and make it short?

Response: Thank you for your critical comments. Comments were taken and revisions have been made accordingly. 

#8: What about your study population of qualitative study?

Response: Thank you for your critical comments. All coordinators of CCA screening unit, senior gynecologist and non-selected female health workers

#9: Why age 21 and above why not age 18?

Response: Thank you for your critical comments. WHO recommends CCA screen for women age 21 years old and above.

#10: exclusion criteria regarding hysterectomy, critical illness, maternal leave…

Response: Thank you for your critical comments. Comments were taken and revisions have been made accordingly. 

 We considered all female health workers age 21 and above from payroll of human resource. After taking each samples, Critical ill women who were unable to communicate and on maternal leave were excluded during study time.

#11: Nothing is presented regarding to qualitative sample size?

Response: Thank you for your critical comments. We do not determine the sample size for qualitative interview rather we depend on the saturation of idea. We conducted nine in-depth interview for purposive selected individuals and those individuals were not interviewed for quantitative arm. Collecting data were stopped when in-depth interviews participants fail to provide new information

#12: is it appropriate definition? What is WHO DEFINITIONS?

Response: Thank you for your critical comments. Correction was made . 

The WHO definition will say: At a minimum, screening is recommended for every woman 30–49 years of age at least once in a life time. American obstetrics and gynecology recommended that the initiation of screening with pap smear at the age of 21-29 years every three years interval followed by screening with pap smear and HPV testing every five years until the age of 65

For our operational definition we used: utilization of CCA screening service: women at least once screened for Cervical cancer in their life time 

Reviewer 2

First, as a general comment, you should get the whole manuscript edited by someone with English as the first language. The current version has significant issues with the use of past and present tenses, the absence of punctuation marks here needed. I have pointed some out below but get this fixed throughout the manuscript. 

Response: Thank you very much for your critical comments. Comments were taken and revisions have been made accordingly.

Specific comments

Introduction

#1.Line 14: change “portion” to proportion and also you write that the women “had” low CC screening. My concern is with the use of the word “had” in the past tense without reference to a specific year or time. I suggest you write that they have low CC screening…

Response: Thank you very much for your critical comments. Comments were taken and revisions have been made accordingly.

#2.Line 17: health workers role in promotion positive health seeking behaviour including CC screening. The current statement suggest that their role is specific to CC screening

Response: Thanks. Definitely, the current role of health profession is specific to CC screening. However, when the community have low awareness about the service availability the have responsibility to make a mass mobilization.

Health-care providers in hospitals and primary health facilities, including community health nurses (CHNs), constitute the most visible, front-line personnel providing health education to patients and the general population. Since CHNs play an integral role in educating women in the prevention of diseases, e.g. in antenatal services and child welfare clinics, they can influence cervical cancer screening adherence and health promotion among women [Krings A, Dunyo P, Pesic A, et al. Characterization of human papillomavirus prevalence and risk factors to guide cervical cancer screening in the North Tongu District, Ghana. PLoS One. 2019;14:e0218762.].

#3.Lines 75-77: the sentence does not end in a meaningful way so do rephrase. 

Response: Thank you very much for your critical comments. Comments were taken and revisions have been made accordingly.

#4.Lines 82-84: again, another sentence that is confusing. To check and rephrase. It currently state “However, none of these studies have assessed the association between the convenience of screening time and the barrier to utilizing cervical cancer screening services were not qualitatively captured”.

Response: Thank you. Comments were taken and revisions have been made accordingly.

#5.Line 84: to state that you aim to determine the “prevalence” in this context is not the most appropriate term. You could say you are determining service utilisation rate or something like that so please rephrase

Response: Thank you. Comments were taken and revisions have been made accordingly.

#6. Lines 86-89: this sentence should come just before the study aim. 

Response: Thank you. Rearrangements have been made accordingly.

#7. Lines 100 – 104: rephrase and made the sentence clear and concise.

Response: Thank you. Comments were taken and revisions have been made.

#8. You need to provide more details on the study recruitment process and data collection. 

o How exactly were the HCWs recruited for the quantitative arm? 

Response: Thank you for your valuable comments. For quantitative study we used simple random sampling technique was used to recruit study participants. In Hossana town, four public health facilities (one comprehensive specialized hospital & three health centers). 

In the first stage, a sampling frame was prepared from the payroll of the human resources department in each four public health facility in Hosanna town to determine the total numbers of female health workers. In the second stage, a total of 241 female workers were selected with probability proportion to the size of each health facility Finally, the study participants were recruited using a simple random sampling method from each facility. (Fig 1).

o Was this a concurrent mixed method, or sequential or what? 

Response: Thank you. We used a concurrent mixed method in which the implementation of qualitative and quantitative methods and merging the data to interpret the study results.

o How were those HCWs recruited for the interviews for the qualitative arm?

Response: Thank you for your concern. We did not used female health workers interviewed for quantitative arm. We have used focal person in screening unit, unselected workers, and senior obstetrician/ gynecologist.

o Who conducted the interviews? 

o Where, for how long? Etc. etc. 

o Since these were HCWs why were the interviews not conducted in English? What measures were in place to ensure that the translation process, did not affect the content of what was said? Was member checking done after transcription?

Response: Thank you for your critical comments. The interviews were completed at a time and place to suit the participants and lasted between 20 and 45 minutes. Memos were used to understand contexts during interview. Categorizing and theming procedures were cross-checked by the other investigators and agreed on common categories and themes. We invited an expert to put sample of codes and categories to the emerged corresponding categories and themes for triangulation, respectively.

o Could someone who responded to the quantitative survey also be in the qualitative interviews? 

Response: No

o What informed the sample size for the qualitative interview? 

Response: Thank you. We do not determine the sample size for qualitative interview rather we depend on the saturation of idea. We conducted nine in-depth interview for purposive selected individuals and those individuals were not interviewed for quantitative arm. Collecting data were stopped when in-depth interviews participants fail to provide new information

#9. Insert the ethical approval reference number.

Response: Thank you. I have included the reference number as IRB/ 153/13

Result part

#10.Why was the age ranges done in this manner? Why not start with the youngest age and then create the ranges to be meaningful? 

Response: Thank you. We have used previous literature to classify our age group. 

#11.Lines 250-251: what does “hugeness and hardiness” mean? If you say this was verbatim, I am wondering what the person said that is translated as hugeness and hardiness. It makes the sentence confusing to the reader. Since the statements after that by the same person sounds clearer, I suggest you delete this first sentence. In fact, the quotations all sound as if the people have low educational level but these are midwives and other level of trained health professionals. I believe the problem is trying to do direct translation from the local dialect to English without giving meaning to the statements. This does not help in this case to make the manuscript read well. Take a look again at your approach to the qualitative data analysis and presentation.

Response: Thank you very much for your critical comments. Comments were taken and revisions have been made accordingly. 

#12.Rephrase the sentence at line 296 and 299 about CC service utilisation. 

Response: Thank you very much for your critical comments. Comments were taken and revisions have been made accordingly. The most common barriers for cervical cancer screening utilization in health facilities were study participants assume themselves as healthy (54.8%) , do not know the place of service (11.5%) , and carelessness (11.1%) on screening method. Figure 2 Barriers for cervical cancer screening utilizations among female health workers in Hossana Town, Southern Ethiopia, 2021

#13.Lines 301-304: this whole part needs to be rephrased. The English language use is poor. This really needs to be addressed throughout the manuscript

Response : Thank you for the comments, corrected.

Two IDIs interviewees pointed out many barriers that may prevent female health workers from using screening services because of assuming a healthy; do not know where service is given in their facility, fear of pain of screening procedure, and being Carelessness as illustrated below:

#14.You repeat to many quotations about the level of CC knowledge, so you must review this and delete some to shorten the manuscript and also make it less boring to the reader.

Response : Thank you for the comments, corrected.

#15. Line 393 has no meaning now. Rephrase because I can’t think through eve to understand what you intend to say

Response: Thank you for the comments, revision was made

#16. Line 395-396: what is lower? Are you still talking about the utilisation rate? If so, be explicit. 

Response: Thank you for the comments, revision was made

The utilization cervical cancer screening service was 19.6% (95% CI: (14.5%, 24.7%). Based on this CI we have discussed as low, comparable and higher.

Discussion and conclusions

#17. These are also full of grammatical errors and these needs to be dealt with

Response: Thank you for the comments, corrected and language edition was made

#18. And then it should expand on recommendations which can address the barriers identified in this study.

Response: Thank you. Comments were taken and revisions have been made accordingly.

Based on the findings of our study we recommend concerned bodies in the health sector to design strategic efforts aimed at improving female health work’s knowledge on cervical cancer and maintain service availability, and . It is also highly valuable to consider upgrading their educational level to realize improved screening service uptake among the groups. Moreover, contextualized health talks and promotion on the availability of screening services and maintain logistics are critical.

---

## [Editor Report · Decision Letter 1]

12 May 2023

Cervical cancer screening utilization and associated factors among female health workers in public health facilities of Hossana town, southern Ethiopia: A mixed method approach

PONE-D-22-33914R1

Dear Dr. Hassen,

We’re pleased to inform you that your manuscript has been judged scientifically suitable for publication and will be formally accepted for publication once it meets all outstanding technical requirements.

Kind regards,

Gedefaye Nibret Mihretie, MSc

Academic Editor

PLOS ONE
---

## [Editor Report · Acceptance letter]

19 May 2023

PONE-D-22-33914R1 

Cervical cancer screening utilization and associated factors among female health workers in public health facilities of Hossana town, southern Ethiopia: A mixed method approach 

Dear Dr. Hasen:

I'm pleased to inform you that your manuscript has been deemed suitable for publication in PLOS ONE. Congratulations! Your manuscript is now with our production department. 

Kind regards, 

on behalf of

Mr. Gedefaye Nibret Mihretie 

Academic Editor

PLOS ONE